# Understanding Internal and External Drivers Influencing the Prescribing Behaviour of Informal Healthcare Providers with Emphasis on Antibiotics in Rural India: A Qualitative Study

**DOI:** 10.3390/antibiotics11040459

**Published:** 2022-03-29

**Authors:** Shweta Khare, Ashish Pathak, Cecilia Stålsby Lundborg, Vishal Diwan, Salla Atkins

**Affiliations:** 1Health Systems and Policy (HSP): Medicines, Focusing Antibiotics, Department of Global Public Health, Karolinska Institutet, Tomtebodavagen 18A, 171 77 Stockholm, Sweden; cecilia.stalsby.lundborg@ki.se (C.S.L.); vishal.diwan@icmr.gov.in (V.D.); 2Department of Public Health and Environment, Ruxmaniben Deepchand Gardi Medical College, Ujjain 456006, Madhya Pradesh, India; 3Department of Pediatrics, Ruxmaniben Deepchand Gardi Medical College, Ujjain 456006, Madhya Pradesh, India; 4International Maternal and Child Health Unit, Department of Women and Children’s Health, Uppsala University, 751 85 Uppsala, Sweden; 5Division of Environmental Monitoring and Exposure Assessment (Water and Soil), ICMR—National Institute for Research in Environmental Health, Bhopal 462030, Madhya Pradesh, India; 6Social Medicine, Infectious Diseases and Migration (SIM), Department of Global Public Health, Karolinska Institutet, Tomtebodavagen 18A, 171 77 Stockholm, Sweden; salla.atkins@tuni.fi; 7Global Health and Development, Faculty of Social Sciences, Tampere University, Arvo Ylpön Katu 34, 33520 Tampere, Finland

**Keywords:** healthcare providers, infectious diseases, caregivers, child, prescription, antibiotics, antibiotic resistance, rural population, India

## Abstract

Globally, Antibiotic resistance is a major public health concern, with antibiotic use contributing significantly. Targeting informal healthcare providers (IHCPs) is important to achieve universal health coverage and effective antibiotic stewardship in resource-constrained settings. We, therefore, aimed to analyse the internal and external drivers that influence IHCPs’ prescribing behaviour for common illnesses in children under five, with an emphasis on antibiotic use in rural areas of India. A total of 48 IHCPs participated in focus group discussions. Thematic framework analysis with an inductive approach was used, and findings were collated in the theoretical framework based on knowledge, attitude, and practice model which depicted that the decisions made by IHCPs while prescribing antibiotics are complex and influenced by a variety of external and internal drivers. IHCPs’ internal drivers included the misconception that it is impossible to treat a patient without antibiotics and that antibiotics increase the effectiveness of other drugs and cure patients faster in order to retain them. Formal healthcare providers were the IHCPs’ sources of information, which influences their antibiotic prescribing. We found when it comes to seeking healthcare in rural areas, the factors that influence their choice include ‘rapid cure’, ‘cost of treatment’, ‘distance’ and ‘24 h availability’, instead of qualification, which may create pressure for IHCPs to provide a quick fix. Targeted and coordinated efforts at all levels will be needed to change the antibiotic prescribing practices of IHCPs with a focus on behaviour change and to help resolve misconceptions about antibiotics.

## 1. Introduction

Antibiotic resistance is a multi-sectoral public health risk on a global scale, posing a threat to patient safety worldwide [1]. Between 2000 and 2015, global antibiotic use increased by 65%, from 21.1 billion to 34.8 billion defined daily doses (DDDs) [2]. According to the report released by the World Health Organisation (WHO) in 2014, we are entering a ‘post-antibiotic’ era, in which antibiotics may no longer be effective in treating illnesses or injuries that were previously treatable [3]. Worldwide, nearly 5.4 million children under the age of five die every year, with infectious diseases accounting for approximately half of all such deaths [4]. Among the most common infections in children are acute respiratory tract infections and acute diarrhoea. Both of these are more likely to be caused by viral infections, which do not need to be treated with antibiotics [5,6,7]. On the other hand, antibiotics are frequently prescribed inappropriately and for illnesses that are not warranted, such as viral infections [8,9,10]. When antibiotic use is increased, antibiotic resistance among the bacteria that cause common infections is increased as well [11]. In many low- and middle-income countries with a high burden of infectious diseases [12] and inadequate antibiotic provision (including overuse and misuse) [13], as well as high rates of antibiotic resistance [14], the first and most severe consequences will be increased healthcare costs, which will have significant social and economic consequences [1]. Each year, an estimated 700,000 people die of diseases caused by drug-resistant pathogens, a figure that might rise to 10 million by 2050 if the current prevalence of drug resistance keeps increasing [15]. Furthermore, by 2050, the annual gross domestic product of the world is expected to decline by 3.8%. Poorer countries will be hit the hardest by the consequences of the greater prevalence of infectious diseases and the greater declines in economic growth [16].

The Indian government took action in April 2017 to combat antibiotic resistance and enacted the National Action Plan on Antimicrobial Resistance, following the approach taken by the WHO in drafting a global antimicrobial resistance action plan. The National Action Plan has numerous goals, including reducing the number of infections, strengthening surveillance, encouraging research, and improving the judicious use of antibiotics [17]. In India, antibiotics sales are regulated by the Drugs and Cosmetics Act and Rules, which provide a mandate for the identification of illegal pharmacies, unqualified doctors and regulate over-the-counter sales of antibiotics [18]. Despite the fact that these provisions exist, informal healthcare providers (IHCPs) can be found practicing in rural areas and are not forced to close down, as it is a well-known fact even to the governing authorities that they address the needs unmet by the formal healthcare system [19]. In this study, the term ‘IHCP’ is used for unqualified medical practitioners. IHCPs are healthcare professionals who have not acquired a formal medical degree from an institution recognised by the Medical Council of India and who are not registered with regulatory body Medical Council of India/State Medical Council(s) as healthcare practitioners [20]. Additionally, anyone qualified in another system of medicine (for example, Ayurveda, Unani, Siddha, or Homoeopathy) who practices any sort of modern system of medicine is categorised under IHCP [21]. The data from a longitudinal follow-up of caregivers’ healthcare-seeking behaviour from rural areas in Ujjain, India, shows that IHCPs were first-contact providers in 73% of cases and provided 85% of the total antibiotics (mostly broad-spectrum) reported during the study [22]. In our previous study, a survey of 15,322 prescriptions given to outpatients by the IHCPs revealed that antibiotics were the most commonly prescribed drug than other medications for common illnesses in both children and adults [8].

IHCPs play important roles in rural healthcare, and their role cannot be overlooked in the effort to achieve Universal Health Coverage (UHC) and designing effective antimicrobial stewardship, especially in resource-limited settings. For them to be more efficiently involved in the provision of healthcare services, it is important to gain an understanding of their knowledge and practice—including how they interpret adequate care and rational treatment. Our aim was, therefore, to analyse the internal and external drivers that influence IHCPs prescribing behaviour for common illnesses in under-five (U-5) children, with an emphasis on antibiotic use, in rural areas of India.

## 2. Results

Our findings suggest that antibiotic prescribing decisions by IHCPs are complex and influenced by multiple drivers at the knowledge, attitude, and practice level. We present our themes grouped into internal and external drivers below, supported by illustrative quotes.

According to our study findings, both internal and external drivers influence IHCPs’ practice, including patient behaviour and demand, availability and access to diagnostic testing, information exchange from formal healthcare providers, and referral facility and sociodemographic characteristics of the IHCPs. In order to give an illustrated overview of the internal and external factors that influence IHCPs prescription behaviours with emphasis on antibiotic prescribing, we compiled our data into a model known as knowledge, attitude, and practices (KAP), proposed by Rodrigues et al.(Figure 1) [23].

### 2.1. Internal Drivers

#### 2.1.1. Theme 1—IHCPs’ Accelerated Therapeutic Interactions with Ready Supply Antibiotics Allowed Them to Surpass Academic Credentials

IHCPs with a long history of providing care are well established and hold a decent reputation among members of their communities. There is hardly any rural community without IHCPs. They are among the vital components of the community since they tend to be less professionally distant and more social by being more interactive with the patients, always having an available source of medicine for their patients, and also adding some personal component to their approach to providing healthcare. These IHCPs and their practices are strongly supported by the villagers. Therefore, IHCPs’ are the major source of healthcare and antibiotics for residents of rural areas.

‘*Like we are practicing and government try to stop us from practicing. Government officials come to the village. So villagers meet head-on with them. They will not let us stop practicing as long as the government doesn’t appoint good doctors or open a government hospital in the village. Also, said that these people will work for us until and unless the government makes other arrangements*’.(Focus Group Discussion (FGD-4); Male 4)

IHCPs have been perceived as a better choice, as they were believed to accelerate therapeutic interactions. As they were primarily local to the patients, they could be more promptly available 24 h, could work a more flexible schedule, and had a readily available supply of medicines and antibiotics. Treatments were expedited as a result of these situations, so IHCPs were seen as a preferable option. Patients who arrived late at night were given antibiotics without hesitation, believing that they were a vital element of the treatment.

‘*Patient came at night and obviously, he cannot go to town at that hour so we give him primary treatment like paracetamol for fever and antibiotics for the night and tell him to go to the paediatrician in the morning*’.(FGD-3; Male 3)

Patients’ paying capacity is taken into consideration by healthcare professionals working in the informal sector, which in turn affects the selection of antibiotics and their doses, resulting in a large number of inadequate doses being administered. IHCPs were supposed to provide more than just good healthcare. Low-income patients benefit from their low-cost therapy and flexible payment alternatives, which reduces their financial burden while enhancing accessibility.

‘*We write 3 days antibiotic dose properly. Actually, in rural areas, it should be at least 5–7 days dose. If we prescribe for complete 7 days, he will find it costly and so he will not take the dose*’.(FGD-2; Female 3)

Patients might easily receive credit from IHCPs for consultation fees and prescription medicines. On occasions, they would even take produce in exchange for money. People in rural areas, many of whom are employed seasonally in agriculture and farming, sometimes do not have the money to pay for healthcare. This was a distinct benefit for IHCPs, as their fees were significantly lower than those charged by traditional healthcare providers.

#### 2.1.2. Theme 2—Beliefs Regarding Antibiotics as a Quick Fix and First Choice in ‘Hit-and-Try’ Prescriptions Increase Antibiotic Use

It appears that IHCPs believe antibiotics as a mandatory part of the treatment. They regarded antibiotics as vital for primary care because they believed antibiotics could heal the majority of diseases. The most common perception was that antibiotics are not for any specific disease and that it is impossible to treat a patient without antibiotics.

‘*Antibiotic is a sure-shot drug. It can cure any disease. Any kind of disease, the patient doesn’t get well without antibiotics. Even if fever is there antibiotic is compulsory, for cough and a cold antibiotic is compulsory, for wounds antibiotic is compulsory, any infection of the body, immunity is enhanced by antibiotics, it is necessary for any disease*’.(FGD-4; Male 4)

Other IHCPs reiterated that antibiotics increase the effectiveness of other drugs if given along with them and enhance the immunity of the body. They also mentioned that antibiotics are already there in the human body:

‘*I mean it boosts up the body, other drugs start working in the body. They are not effective if antibiotics aren’t given. If antibiotics are not included in the treatment it takes a very long time to get better and many a time there is no effect at all*’.(FGD-5; Male 4)

Another perception of some IHCPs was that more antibiotics cure faster and if any antibiotic resistance is at all present, at least one of the antibiotics in the combination or one of the antibiotics among the two or three prescribed will work:

‘*Knowledge also and they also want fast results. Like they are giving a combination of 2 antibiotics, whether it is needed or not, so accordingly he is adding and giving believing that the patient will get well fast. The patient might not get cured by giving only cefixime, so cloxacillin in combination is given or something else and he thinks that patient will get fine because cases of resistance are also coming as patients are not taking full dose*’.(FGD-4; Male 4)

One more reason for practicing more antibiotic prescribing, as mentioned by one of the IHCP, was to retain the patients. In the opinion of IHCPs, patients wanted the fastest cure possible; otherwise, they would go to other providers; hence, antibiotics were deemed absolutely necessary as part of the treatment plan:

‘*Also happens when the provider is having competition and do not want to lose his patients to another provider thinking that if the single antibiotic prescribed did not affect and the patient does not get better soon he will go to another provider and so they prescribe more than one antibiotic*’.(FGD-7; Male 1)

Antibiotics, they reasoned, were necessary for primary care since they could treat the majority of infections. The decision to use an antibiotic is not based on scientific evidence. Antibiotic selection and dose were based on IHCPs’ personal experiences with the effectiveness of various antibiotics for certain illnesses. The ‘hit-and-try’ approach is used to choose antibiotics for a certain disease. Over time, they’ve come to realise which antibiotics to administer for which symptoms and diseases.

‘*Prescription of antibiotics, we do it from our experience like if the patient has a fever, in a normal way we give simple medicines many times, the fever didn’t subside then second-time small antibiotic is given, Amoxicillin, MOX syrup with the fever and so we got the result. Madam, we learnt with our experience that in fever if one antibiotic is given with antipyretic then the body will get relief faster. Yes, experience teaches everything, we keep track that this was the condition, and for this condition, this medicine was better*’.(FGD-6; Male 3)

Knowledge of the cause and development of antibiotic resistance varies widely among the IHCPs. Many of the IHCPs did not know about antibiotic resistance. Some of the IHCPs identified incorrect duration and incorrect dose of the drug as causes of occurrence of antibiotic resistance.

‘*They get treatment of cough cold in the village for 2 days, but they will take treatment for only 1 day, we explain them properly, that it is very necessary to take antibiotics for 3 days and sometimes for 5 days if needed if you don’t take the complete course resistance will develop and it will stop acting on you*’.(FGD-5; Male 4)

Furthermore, only a small number of IHCPs recognise that antibiotic resistance is a result of incorrect intake and that they themselves may be at fault. They justified this by stating that they do not have a thorough understanding of how and when to administer antibiotics.

‘*We are not educated in medicine; it is possible that we could be the reason for the resistance*’.(FGD-4; Male 6)

### 2.2. External Drivers

#### 2.2.1. Theme 3—Mutually Beneficial Relationship between Informal and Formal Healthcare Providers Led to Available Antibiotics

The study results revealed that there was no knowledge of the different types of common acute illnesses in U-5 children among IHCPs. Their understanding of the causative organism was limited to simple inferences such as that indicated in the following statement:

‘*According to my experience, the tiny beings which we can’t see from our naked eyes, they are bacteria. The ones which are tinier than bacteria, they are viruses which are dead when outside and when they reach inside the body they become alive*’.(FGD-6; Male 2)

Formal healthcare providers are the IHCPs’ source of information on the treatment of illnesses which thus influence their antibiotic prescribing. IHCPs gather knowledge by working as assistants under formal healthcare providers in their clinics or pharmacies, while some have also taken one-time certificate courses organised by the government. Most of the practice of the IHCPs is based on the learning gained by studying and following the prescriptions written by formal healthcare providers or by consulting these formal healthcare providers, and in return, when needed, they refer their patients to them.

‘*I worked at the medical retail counter for 11 years so I have the knowledge of medicines and from time to time we get training from the civil hospital, welfare society, Pushpa Mission Hospital, etc. Apart from this whenever needed we take advice from child specialists*’.(FGD-3; Male 4)

‘*Madam, this we have learned during ‘Jan sawasthya rakshak’ (Community Health Visitor) training. Also, doctors give lectures to us and tell us to come to see patients, make groups of five, and take one group every day. To study what he had diagnosed and written on a treatment pad. You have to stand there and observe*’.(FGD-2; Male 1)

#### 2.2.2. Theme 4—Patients Thought That Antibiotics Were Effective and Often Demanded Them, Leading to Prescriptions

IHCPs who participated in FGDs had varying perspectives on patients’ knowledge of antibiotics; infact, it is possible that some patients have never ever heard of them; hence, there was little evidence of patients demanding antibiotics while sick. However, the majority of patients need rapid cure owing to the insecure nature of their daily wage jobs, and at a reasonable cost, which translates into a shorter course of antibiotic treatment. They relied heavily on IHCPs for healthcare and antibiotics. Additionally, several other IHCPs remarked that there are demanding patients who are difficult to convince, show dissatisfaction when they do not obtain what they desire, and frequently shift to other healthcare providers. As a result, IHCPs’ decisions about whether or not to administer antibiotics are impacted. They feared that if they did not prescribe antibiotics, patients would not recover, and they would lose their patients.

‘*In village what happens, labour class are there, they don’t have time, they come in a hurry that they have to go back for their work, for labour work, the kid should be alright instantly; if he doesn’t get relief in next 2 h, they will come to me, we both are from the same village, if I am unable to give relief, they will go to him; if he is not able to provide the relief, they will go to someone else. But the kid should get relief in 1 h. Take treatment here for one day, if it gets alright, then it is ok; otherwise, we have to go to Ujjain or show to the big doctor. In such situations, we are also not able to make a decision that what type of medicine should be given*’.(FGD-2; Female 5)

IHCPs noted that incredibly, some patients were able to name certain antibiotics even though they had no idea how or why they had been provided with these medications/antibiotics. This is because many patients in rural areas have been accustomed to the instant relief of symptoms with antibiotics obtained from a medical shop, without fully comprehending the hazards or uses of these medications. We also discovered that patients put pressure on their doctors to give injectables instead of oral drugs because they believe that injectables work faster.

‘*They purchase from medical, by directly saying give us cefixime, give us monoxil, amoxicillin. Just like zandu balm (Ayurvedic ointment for pain relief), the head is aching so give us zandu balm. They take antibiotics like that which they have learned from medical stores. If anybody goes to the store and tell them their symptoms, the medical storekeeper will make a complete dose for them and give and tell them that this is antibiotic, this is for fever, and this is for cough cold, so this is how they learn*’.(FGD-4; Male 4)

## 3. Discussion

We identified key internal and external drivers and interconnection among drivers influencing the prescribing behaviour of IHCPs, with emphasis on antibiotic prescribing for common illnesses in U-5 children in rural areas in central India, in order to guide the implementation of a customised social marketing intervention. The study results also reflect that despite the lack of official recognition, IHCPs are part of wider rural social systems and settings, and their prescription practices are influenced by all of the different parts of this system. A KAP model was used to visualise and summarise these factors in order to better understand the complexities of prescribing behaviour of IHCPs. Such models can help to investigate and determine the healthcare providers’ subjective views or the type of healthcare providers’ related attitudes and knowledge [24]. While other studies have reported similar findings [24,25,26], our findings aligned with a recent study of antibiotic prescribing by IHCPs in rural West Bengal, India, which concluded that IHCPs must be considered as an essential part of the rural healthcare system for developing an effective antibiotic stewardship strategy and that multiple stakeholders must be targeted with a variety of regulatory, educational, and behaviour change interventions [27]. Highlighting the significance of this research even further is the fact that there is no qualitative study from this region that demonstrates the interconnection between internal and external drivers influencing the prescribing behaviour of IHCPs, especially antibiotics. Discussed below are the key findings and potential determinants for effective interventions promoting prudent antibiotic prescription in rural India and in similar contexts.

### 3.1. IHCPs Sociodemographic Characteristics, Awareness, Knowledge, and Misconceptions Regarding Antibiotic Use

The National Health Mission, which was implemented in 2005, envisioned the use of accredited social health activists (ASHAs) as the first point of contact for community health-related problems in order to strengthen the existing three-tier rural public health delivery system [28]. The results of our study, however, suggest that IHCPs continue to function as the initial point of contact for primary level curative healthcare and play important roles in providing healthcare and antibiotics in rural areas. Our analysis supported the findings of numerous earlier studies showing that IHCPs provide a large portion of rural populations’ basic health care [29,30,31,32,33]. Our study results show that, among the rural population, it was not a matter of qualification of healthcare providers; when it comes to seeking healthcare, the factors which affects their selection were ‘rapid cure’, ‘cost of treatment’, ‘distance’, and ‘24 h availability’, which may create pressure for IHCPs to provide a quick fix. Other comparable research also found that proximity and round-the-clock availability were the primary motivators for accessing IHCPs [33,34]. People in rural areas seek primary care that is not doctor-centric and is not given at a distant doctor’s office [30].

Our study findings show that prescribing by the IHCPs was influenced by a combination of misperceptions about antibiotics as a medical requirement. This outcome is consistent with previous research indicating that IHCPs believe antibiotics are critical for patient retention and financial stability [34]. IHCPs during FGDs in our study stated that they chose antibiotic therapy based on affordability for patients and that they modify antibiotic courses based on the patient’s ability to pay. Additionally, prescribe antibiotics out of fear of losing patients. Affordability, comfort, and trust [35] all also play determining roles in urban poor populations seeking care from IHCPs despite increased availability to skilled health care professionals in urban areas [36]. In order to improve the rural public healthcare system and to increase the uptake of formal healthcare providers, it is imperative that the governing bodies reform and modify the current three-tier system to better meet the demands of patients and their choices.

### 3.2. Influence of the Formal Healthcare Providers

Our research team also studied the prescriptions by IHCPs in the same setting in the different studies and reported that 75% of prescriptions by IHCPs contained antibiotics and were prescribed more frequently than any other medicine [8]. However, this study’s findings suggest that antibiotic selection and dose patterns prescribed by the IHCPs were arbitrary, without following established treatment protocols or regulatory standards. Our study also reported that IHCPs learn from formal doctors by working at their clinics or medical shops or by following the prescriptions written by formal doctors. Therefore, IHCP practice is akin to a spillover impact from the formal system to these IHCPs, a diluted practice. Gautham et.al reported that IHCPs receive information from formal providers in return for the referral of patients to them [27]. Another study reported that the antibiotic prescribing by formal healthcare providers was 55% greater than that of IHCPs [37]; therefore, such guidance by formal providers may be a major factor in this antibiotic misprescription. In a study by our group, it was reported that IHCPs informally learn from formal providers and improve over time as they gain experience. Thus, offering educational opportunities for these IHCPs and integrating them into the official healthcare system might be some of the innovative strategies to serve the larger rural population [38].

Previous studies comparing IHCPs to formal health providers have shown that IHCPs’ knowledge and methods differ, with lack of adequate training highlighted as a significant underlying issue [39]. However, other studies also reported that the quality of service provided is poor, there are knowledge gaps, and conflicting views exist on whether IHCPs should be integrated into the formal healthcare system [32,40,41]. Attracting and retaining formal providers in rural parts of low- and middle-income countries, particularly India, has proven difficult for health ministries, jeopardising rural healthcare, which forms the majority of the country’s population [42]. While IHCPs fill this gap [43], the Medical Council of India still does not want IHCPs to be part of the formal healthcare system. However, the current government of India is dedicated to achieving UHC by focusing on improving the public health system and putting the needs of its citizens first. India’s flagship ‘skill development’ programme provides short-term training or courses to improve the abilities of IHCPs [44]. The Indian government’s ambitious Ayushman Bharat Scheme aims at upgrading rural sub-centres and primary health centres to health and wellness centres staffed by mid-level providers to give primary care, as one way of strengthening rural primary care [43]. However, India’s huge rural population may be underserved by these figures even if they are completely operational. Therefore, in order to fully strengthen the rural public healthcare system and design effective antimicrobial stewardship, private practitioners, including IHCPs, might be included in the UHC initiative [27].

### 3.3. Methodological Considerations

The IHCPs were difficult to convince to participate and share details related to their practice, so the first strength of the study is that we were able to conduct these FGDs with this hard-to-reach group. Another strength of our study stems from the depth of our data. It made it possible for us to present our findings in accordance with the KAP model, where it was demonstrated that antibiotic prescribing behaviour is influenced by a plethora of internal and external drivers at all levels of the model. The FGDs provided extensive insights into IHCPs’ prescribing behaviour. Though most of our respondents were male, and this represents the distribution of IHCPs in this setting, we aimed to include participants of both sexes and years of work experience. There is a possibility of information bias; respondents might be influenced to offer an answer that was either right or positive. This is a preliminary study, and our results’ transferability is limited to areas with similar settings. It is vital to note that our study does not include informal pharmacists, which is critical for future studies. Further research is needed involving multiple stakeholders, such as patients and informal pharmacists, to have a deeper knowledge of individual aspects at all levels of the model. This would aid in a more thorough understanding of and response to the factors that drive antibiotic prescriptions, guiding an effective antibiotic stewardship programme.

## 4. Materials and Methods

### 4.1. Study Setting

The study was conducted in rural areas in the Ujjain District of Madhya Pradesh state, India. A vast, three-tiered rural public healthcare system exists in India, with the goal of ensuring primary care to all individuals regardless of socioeconomic status [45]. The sub-health centre (first tier) is the most remote and initial point of contact. According to population guidelines, one sub-centre is built for every 5000 people in rural areas and 3000 people in hilly/tribal/desert areas and is expected to be staffed by qualified health care professionals and auxiliary nurse midwives [46]. Primary Health Centres (second tier) serve as initial points of contact between village residents and a medical officer. According to population guidelines, one primary health centre is built for every 30,000 people in rural areas and 20,000 people in hilly/tribal/desert areas. They are to be staffed by a physician, paramedics, and other support personnel [47]. Community Health Centre (third tier) serves primarily as a referral centre for the nearby primary health centres. They are supposed to have medical experts (a surgeon, general practitioner, a gynaecologist, and a paediatrician), as well as paramedics and other support personnel, as well as 30 beds and amenities such as an operating theatre and a radiography room [48]. However, this system has not been able to provide effective care, a key reason for the failure of this healthcare system is the lack of trained physicians and nurses willing to work in rural regions, resulting in a shortage [49], as well as the absence of, appointed medical staff for a lengthy period of time [42,50]. Rural India accounts for approximately 71% of the overall population, but only 36% of all healthcare providers work in rural areas [51]. The doctor-population ratio in India is 1:1456, compared with the WHO recommendation of 1:1000, whereas the urban-to-rural doctor population ratio is wildly skewed [43,52]. IHCPs have emerged as a result of the public healthcare system’s inability to deliver [45]. In 2019, 57.3% of individuals practicing allopathic medicine lack a medical degree, which, according to the WHO, is defined as IHCPs [43].

The Ujjain District is located in the Indian state of Madhya Pradesh. It has a population of 1.9 million people, with 61% residing in rural regions [53]. The district has a literacy rate of 72.3% [53] and the poverty rate of about 49%, which is higher than the national average of 21.2% [54]; the maternal mortality rate is 176 per 100,000 live births, and the infant mortality rate is 54 per 100,000 live births [55], both of which are higher than the national average. Both the public and private sectors contribute to Ujjain’s healthcare system. In Ujjain, IHCPs account for 56% of all health care professionals, with the percentage being higher in rural areas [8,56].

### 4.2. Study Design and Participants

A qualitative approach was deemed appropriate to address our study aim. Following a literature search and gathering expert opinions, a discussion topic guide (Appendix A) was developed. It was pilot tested by conducting one FGD with 8 IHCPs in March 2016 and revised accordingly. The discussion guide was first developed in English and then translated into Hindi for its use in the field. Snowball sampling was used for participants’ selection [57] due to the IHCPs’ reluctance to divulge their identities and details. Therefore, the first author (S.K.) contacted the eight IHCPs who had been connected with our medical college (Ruxmaniben Deepchand Gardi Medical College) with the purpose of obtaining information about treatment by attending various workshops and seminars held by the medical college, and they were willing to participate after describing the project aim. Then, the IHCPs in the first sample group were requested to help in contacting other IHCPs among their acquaintances, with the goal of obtaining a varied group of IHCPs in terms of age, gender, education, years of experience, and place of practice. This method of recruiting helped the researcher with the opportunity to communicate better with the participants who otherwise were reluctant to participate [58]. In total, 48 IHCPs took part in 7 FGDs. Age ranged from 19 to 61 years (mean ± standard deviation (SD) of 41 (±6) years); IHCPs experience ranged from 2 to 34 years (mean ± SD of 16 (±5) years) (Table 1).

### 4.3. Data Collection

The first author (S.K.), experienced in qualitative methods, conducted all the FGDs with the assistance of a research assistant skilled in moderating discussions, between December 2016 and February 2017. At the beginning of the FGD, IHCPs were given different case scenarios; for example, if a parent with a child U-5 years of age visits their clinic and complains about frequent watery stools since last night, how would they treat the child’s illness? What treatment would they prescribe, and why would they that treatment? Then, the facilitator used a topic guide to help participants discuss their treatment experiences. The FGDs were conducted in Hindi in which the participants, as well as the data collectors, were conversant. In total, 5–8 people participated in each FGD. After the seventh FGD, we discovered that no new information had emerged, and the sampling was then closed. One FGD was conducted with a group of female IHCPs; six were conducted with groups of male IHCPs. Audio recordings of the FGDs were performed with the participants’ permission, and each discussion lasted for around 60–90 min. In addition, field notes and reflections were taken during the interviews. A light snack was provided to the attendees following the FGDs. Participants in the study were not given any additional incentives.

### 4.4. Data Analysis

Transcription was carried out in Hindi, and the content was translated into English by S.K. and research assistants. S.K. and S.A. (an experienced qualitative researcher) analysed FGD transcripts iteratively, using the thematic framework analysis with an inductive approach [59].Beginning with the first FGD, the process of analysis was initiated, allowing for the integration of results into later focus group work. To begin, transcripts were read independently multiple times and then were coded manually. Certain topics emerged from the literature research; others were identified inductively from the data through the open coding approach. This process was repeated, and finally, similar codes were grouped together and condensed into categories, then grouped into themes (Table 2). Analysis was discussed repeatedly between S.K. and S.A., and later with all co-authors until consensus was reached. Finally, findings were presented in the theoretical framework based on the KAP model [23], to depict how IHCP’s practicing behaviour is influenced by processes and interactions at 3 levels: knowledge, attitude, and practice.

## 5. Conclusions

We identified key drivers under the knowledge, attitude, and practice component of the model, where knowledge to practice was not always unidirectional, and the attitude of the IHCPs often affected their knowledge, as they showed interest in more training, to collect more knowledge, and improve practice. Similarly, the practice also helped to improve their knowledge, as they were untrained in medicine but with experience of working under formal healthcare providers and then with independent practice, they acquired more knowledge. By using a KAP model, we were able to discern the extent of the problem’s complexity and the many implications it has for IHCPs’ decisions and behaviour when it came to prescribing antibiotics. Therefore, these areas can be targeted to successfully improve IHCPs’ prescribing behaviour with emphasis on antibiotics for common illnesses in U-5 children. IHCPs are a vital part of the rural healthcare system. However, they were unable to deliver proper treatment and care due to a lack of training and knowledge. Their treatments were not always consistent with guidelines, and a number of barriers reported in the study prevented them from providing proper treatment. In addition, we found that IHCPs had a limited understanding of antibiotic resistance and the risks of irrational antibiotic use. When establishing a successful antibiotic stewardship strategy, it is critical that regulatory bodies recognise and acknowledge their role in the health system and then lawfully offer different interventions targeting the factors that affect their practices with a combination of regulatory, educational, and behaviour modification interventions. Under the Government of India’s flagship ‘skill development’ programme, planned short-term training and courses to upgrade the skills of IHCPs can focus on improving diagnosis, enhancing IHCPs’ and patients’ understanding of antimicrobial resistance and rational use of antibiotics, and monitoring behavioural changes among these groups, as well as improving the referral of patients from rural areas. In addition, the services provided by IHCPs in rural areas can be supported with incentives. Finally, more study is needed to better understand the opinions and behaviour of the general public and informal pharmacists in order to develop effective intervention strategies that involve all important stakeholders.

## Figures and Tables

**Figure 1 antibiotics-11-00459-f001:**
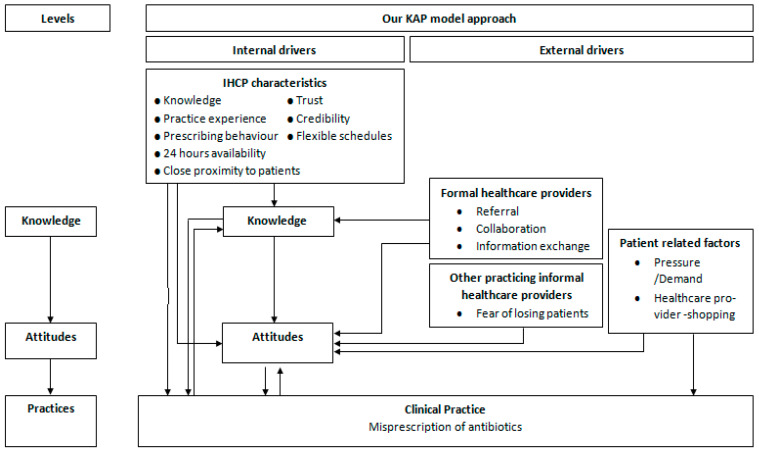
Overview of internal and external drivers identified, with interconnection among the drivers and impact on informal healthcare providers prescribing behaviour at all levels of the KAP model.

**Table 1 antibiotics-11-00459-t001:** Informal healthcare provider’s demographic characteristics (*n* = 48).

Variables	Frequency, n
**Age (years)**	
19–28	1
29–38	7
39–48	37
49–58	2
59–68	1
**Gender**	
Male	43
Female	5
**Education**	
Secondary education (10thgrade)	14
Senior secondary education (11th–12th grade)	16
Higher education(BA/BSc/BCom) *	18
**Experience as informal healthcare provider (years)**	
1–10	6
11–20	36
21–30	5
31–40	1

* BA—Bachelor of Arts, BSc—Bachelor of Science, BCom—Bachelor of Commerce.

**Table 2 antibiotics-11-00459-t002:** Illustrative table of analysis undertaken.

Responses (Meaning Units)	Codes	Categories	Themes
“Like we are practicing and government try to stop us from practicing. Government officials come to village. So villagers meet head-on with them. They will not let us stop practicing as long as government doesn’t appoint good doctors or open a government hospital in the village. Also, said that these people will work for us until and unless government makes other arrangements”. (FGD 4, Male 4)	IHCPs have support from villagers which gives them the confidence to practice.	Perception of informal healthcare providers	IHCPs’ accelerated therapeutic interactions with readily available antibiotics allowed them to surpass academic credentials.
“Patient came at night and obviously, he cannot go to town at that hour so we give him primary treatment like paracetamol for fever and antibiotics for the night and tell him to go to paediatrician in the morning”. (FGD 3, Male 3)	IHCPs are available in the village and are available with medicines such as antibiotics, paracetamol, etc. during the wee hours.	IHCPs are approached for the treatment, as they are available for 24 h	
“They don’t give us any fees; they don’t give us money; they can give 200–400 rs to doctor in city but not to us that is why we do not give injections and prescribe accordingly, see I will say what is reality”. (FGD 1, Male 1)	Patients ask for the treatment for which they can pay or for half of the treatment and so prescribe accordingly.	The financial condition of the caregiver affects the treatment prescribing behaviour of IHCPs	
“Antibiotic is a sure-shot drug. It can cure any disease. Any kind of disease, patient doesn’t get well without antibiotic. Even if fever is there, antibiotic is compulsory, for cough and cold antibiotic is compulsory, for wounds antibiotic is compulsory, any infection of the body, immunity is enhanced by antibiotics, it is necessary for any disease”. (FGD 4, Male 4)	Antibiotics boost the immune system of the body; viral will not get covered without antibiotics.	Knowledge of informal healthcare providers about antibiotic use	Beliefs regarding antibiotics as a quick fix and first choice in ‘hit-and-try’ prescriptions increase antibiotic use.
“I mean it boosts up the body; other drugs start working in the body. They are not effective if antibiotics aren’t given. If antibiotics are not included in the treatment it takes a very long time to get better and many times there is no effect at all”. (FGD 5, Male 4)	Improves the efficiency of other drugs when given along with them.	IHCPs consider antibiotics as the mandatory part of the treatment	
“Prescription of antibiotics, we do it from our experience like if patient has fever, in a normal way we give simple medicines many times, fever didn’t subside then second time, small antibiotic is given, Amoxicillin, MOX syrup with the fever and so we got the result. Brother, we learnt with our experience that in fever if one antibiotic is given with antipyretic, then body will get relief faster. Yes, experience teaches everything; we keep a track that this was the condition and for this condition, this medicine was better”. (FGD 6, Male 3)	Selection of antibiotics for an illness is based on ‘hit-and-trial’ method and learning with experience.	Antibiotics prescribing practice of IHCPs	
“I worked at medical retail counter for 11 years so I have the knowledge of medicines, and time to time, we get training from civil hospital, welfare society, Pushpa Mission Hospital, etc. Apart from this, whenever needed, we take advice from child specialists”. (FGD 3, Male 4)	Gain knowledge about different treatments by assisting other formal practitioners and attending training sessions held by medical institutions.	Learning by observing and attending treatment training sessions	Mutually beneficial relationships between informal and formal healthcare providers led to available antibiotics.
“In village what happens, labour class are there, they don’t have time, they come in a hurry that they have to go back for their work, for labour work, kid should be alright instantly; if he doesn’t get relief in next 2 h, they will come to me, we both are from same village; if I am unable to give relief, they will go to him; if he is not able to provide relief, they will go to someone else. But kid should get relief in 1 h. Take treatment here for one day, if it gets alright then it is ok, otherwise we have to go to Ujjain or show to big doctor. In such situations, we are also not able to take decision that what type of medicine should be given.”. (FGD 2, Female 5)	Caregivers put pressure for quick relief.	Barriers in providing appropriate treatment	Patients thought that antibiotics were effective and often demanded them, leading to prescriptions.

## Data Availability

The data presented in this study are available on request from the authors.

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
