# Peer review of "Understanding Internal and External Drivers Influencing the Prescribing Behaviour of Informal Healthcare Providers with Emphasis on Antibiotics in Rural India: A Qualitative Study"

_antibiotics, 2022, doi:10.3390/antibiotics11040459_

Round 1
Reviewer 1 Report
The Study being qualitative, used appropriate methods and innovatively presented the results in a format that can be understood. However, the abstract can be improved to reflect a summary of the significant results. The conclusion is so general that it cannot guide the required intervention. For Example, what should the regulatory agency do in these informal facilities?
Figure 1 should be edited to ensure that the content fit in the table. No attachment
Author Response
To,
Editor
Antibiotics
Subject: Submission of response to the reviewer_1 comments on the manuscript (Manuscript ID: antibiotics-1611636)
Title: Understanding internal and external drivers influencing the prescribing behaviour of informal healthcare providers with emphasis on antibiotics in rural India: A qualitative study
Dear Editor,
Thank you very much, for the reviewer’s observations and comments on our manuscript. We are herewith submitting point by point reply to comments/observations. Kindly note our reply is in red.
Reviewer 1
Comments
- The study being qualitative used appropriate methods and innovatively presented the results in a format that can be understood. However, the abstract can be improved to reflect a summary of the significant results.
Response 1:
Thank you very much for the suggestion. In response to your above suggestion, we have reworded the result part of the abstract. Now, the results are as follows:
“Thematic framework analysis with an inductive approach was used and findings were collated in the theoretical framework based on knowledge, attitude, practice model which depicted that the decisions made by IHCPs while prescribing antibiotics are complex and influenced by a variety of external and internal drivers. IHCP's internal drivers included the misconception that it is impossible to treat a patient without antibiotics and that antibiotics increase the effectiveness of other drugs and cure patients faster in order to retain them. Formal healthcare providers were the IHCPs’ sources of information, which influences their antibiotic prescribing. We found when it comes to seeking healthcare in rural areas, the factors that influence their choice include "rapid cure," "cost of treatment," "distance" and "24-hour availability," instead of qualification, which may create pressure for IHCPs to provide a quick fix.”
The changes are incorporated in Section: Abstract
- The conclusion is so general that it cannot guide the required intervention. For Example, what should the regulatory agency do in these informal facilities?
Response 2:
Thank you very much for your suggestion. In response to your above suggestion, we have reworded some parts of the conclusion. Now, the conclusion also includes:
“IHCPs are a vital element of the rural healthcare system. However, they were unable to deliver proper treatment and care due to a lack of training and knowledge. Their treatments were not always consistent with guidelines, and a number of barriers reported in the study prevented them from providing proper treatment. In addition, we found that IHCPs had a limited understanding of antibiotic resistance and the risks of irrational antibiotic use. When establishing a successful antibiotic stewardship strategy, it is critical that regulatory bodies recognize and acknowledge their role in the health system and then lawfully offer different interventions targeting the factors that affect their practices with a combination of regulatory, educational, and behaviour modification interventions. Under the Government of India’s flagship "skill development" programme, planned short-term training and courses to upgrade the skills of IHCPs can focus on improving diagnosis, enhancing IHCPs' and patients' understanding of antimicrobial resistance and rational use of antibiotics, and monitoring behavioural changes among these groups, as well as improving the referral of patients from rural areas. In addition, the services provided by IHCPs in rural areas can be supported with incentives.”
The changes are incorporated in Section: Conclusion
- Figure 1 should be edited to ensure that the content fits in the table.
Response 3:
We've included the corrected figure beneath the previously submitted figure because we were having difficulty modifying it appropriately. Now, all of the figure's components are visible.
The changes are incorporated in Section: Results
Yours Sincerely
Dr Shweta Khare
Department of Global Public Health
Karolinska Institutet
Email: shweta.khare@ki.se
Mar 16 2022

Reviewer 2 Report
This research investigates the drivers of attitude of prescribing antimicrobials by non-official medical practitioners, who actually represent a major share of healthcare providers in rural India.
The topic is of great interest and the methods are adequate and sound, as well as the conclusions are coherent with the evidences, which are adequately discussed, although the scientific background could be better represented.
However, I think there are some parts that can be improved.
Overall, the introduction could be expanded, especially from lines 42 to 51 trying to provide a broader overview.
Line 44: Consider using the symbol % instead of “per cent”.
Line 45: Consider writing "by the World Health Organization (WHO)".
Line 51: I recommend a more deep analysis of antimicrobial resistance. Consider expanding the introduction by describing the novel research about alternative therapies in order to contrast antibiotic resistance.
For instance, consider this article:
Troiano G, Messina G, Nante N. Bacterial lysates (OM-85 BV): a cost-effective proposal in order to contrast antibiotic
resistance. J Prev Med Hyg 2021;62:E584-E573. https://doi.org/10.15167/2421-4248/jpmh2021.62.2.1734
Line 96 to 110: The figure is misplaced in the layout, and thereby unreadable.
Line 276: “While other studies have…” There is no bibliographic reference to support the statement.
Line 313: Assuming that the subject of this sentence, “we”, are indian healthcare providers, I recommend not speaking in 1st person if not referencing to the authors.
Lines 353 to 365: Please discuss further the limits of the study. Should this be considered as a preliminary study and how much this results are viable for planning interventions?
Line 404: I recommend explaining the meaning of the acronym "FGD" at its first occurrence.
Line 408: Author contributions are already included in the paragraph "Author contribution", you could replace “So, the first author…”with "We contacted...". Do the same where you used the same methodology, for example line 422
Line 450: Since the acronym "KAP" appears the first time at line 94, I recommend explainingbetter and the methodology.
The article number 46 has already been published, the wording “in press” is incorrect.
For this reasons, I suggest a major revision.
Please, view the comments inside the PDF file attached.

Author Response
To,
Editor
Antibiotics
Subject: Submission of response to the reviewer_2 comments on the manuscript (Manuscript ID: antibiotics-1611636)
Title: Understanding internal and external drivers influencing the prescribing behaviour of informal healthcare providers with emphasis on antibiotics in rural India: A qualitative study
Dear Editor,
Thank you very much, for the reviewer’s observations and comments on our manuscript. We are herewith submitting point by point reply to comments/observations. Kindly note our reply is in red.
Reviewer 2
Comments
- This research investigates the drivers of attitude of prescribing antimicrobials by non-official medical practitioners, who actually represent a major share of healthcare providers in rural India. The topic is of great interest and the methods are adequate and sound, as well as the conclusion is coherent with the evidences, which are adequately discussed, although the scientific background could be better represented. However, I think there are some parts that can be improved. Overall, the introduction could be expanded, especially from lines 42 to 51 trying to provide a broader overview.
Line 51: I recommend a more deep analysis of antimicrobial resistance. Consider expanding the introduction by describing the novel research about alternative therapies in order to contrast antibiotic resistance.
For instance, consider this article: Troiano G, Messina G, Nante N. Bacterial lysates (OM-85 BV): a cost-effective proposal in order to contrast antibiotic resistance. J Prev Med Hyg 2021;62:E584-E573. https://doi.org/10.15167/2421-4248/jpmh2021.62.2.1734
Response 1:
Thank you for your suggestion. We have added literature to the introduction describing more about the development of antibiotic resistance. However, novel research to contrast antibiotic resistance was not the objective of the study, so we were not able to relate and put that component into the introduction.
The part of the introduction read as follows-
Antibiotic resistance is a multi-sectoral public health risk on a global scale, posing a threat to patient safety worldwide [1]. Between 2000 and 2015, global antibiotic use increased by 65 %, from 21.1 billion to 34.8 billion defined daily doses (DDDs) [2]. According to the report released by the World Health Organization (WHO) in 2014, we are entering a "post-antibiotic" era, in which antibiotics may no longer be effective in treating illnesses or injuries that were previously treatable [3]. Worldwide, nearly 5.4 million children under the age of five die every year, with infectious diseases accounting for approximately half of all such deaths [4]. Among the most common infections in children are acute respiratory tract infections (RTIs) and acute diarrhoea. Both of these are more likely to be caused by viral infections, which don't need to be treated with antibiotics [5-7]. On the other hand, antibiotics are frequently prescribed inappropriately and for illnesses that are not warranted, such as viral infections [8-10]. When antibiotic use is increased, antibiotic resistance among the bacteria that cause common infections is increased as well [11]. In many low- and middle-income countries (LMICs) with a high burden of infectious diseases [12] and inadequate antibiotic provision (including overuse and misuse) [13], as well as high rates of antibiotic resistance [14], the first and most severe consequences will be increased healthcare costs, which will have significant social and economic consequences [15]. Each year, an estimated 700,000 people die of diseases caused by drug-resistant pathogens, a figure that might rise to ten million by 2050 if the current prevalence of drug resistance keeps increasing [16]. Furthermore, by 2050, the annual gross domestic product of the world is expected to decline by 3.8%. Poorer countries will be hit the hardest by the consequences of the greater prevalence of infectious diseases and the greater declines in economic growth [17].
The changes are incorporated in the section: Introduction
- Line 44: Consider using the symbol % instead of “per cent”
Response 2:
Thank you for the suggestion the relevant changes have been incorporated at the respective places.
- Line 45: Consider writing "by the World Health Organization (WHO)".
Response 3:
Thank you for the suggestion the relevant changes have been incorporated at the respective places.
- Line 96 to 110: The figure is misplaced in the layout, and thereby unreadable.
Response 4:
We've included the corrected figure beneath the previously submitted figure because we were having difficulty modifying it appropriately. Now, all of the figure's components are visible.
The change is incorporated in the section: Results
- Line 276: “While other studies have…” There is no bibliographic reference to support the statement.
Response 5:
Thank you for noting it down, references have been added at the respective place.
- Line 313: Assuming that the subject of this sentence, “we”, are Indian healthcare providers, I recommend not speaking in 1st person if not referencing to the authors.
Response 6:
Thank you for noting it down, now the sentence reads as follows-
“In order to improve the rural public healthcare system and to increase the uptake of formal healthcare providers, it is imperative that the governing body reform and modify the current three-tier system to better meet the demands of patients and their choices.”
- Lines 353 to 365: Please discuss further the limits of the study. Should this be considered as a preliminary study and how much these result are viable for planning interventions?
Response 7:
Thank you for your suggestion. In response to the reviewer’s comment, we would like to inform you that yes this is a preliminary study but is a part of a project where we have investigated the ‘One health’ issues regarding antibiotic use and antibiotic resistance in children and their environment in Indian villages. We have studied parents’ healthcare-seeking behaviour for their children (1–3 years of age at the onset), prescribing patterns of formal and informal healthcare providers, analysis of phenotypic antibiotic resistance of Escherichia coli from samples of stool from children and village animals, household drinking water, village source water and wastewater, and investigation on molecular mechanisms governing resistance. So, the future plan is to design an intervention taking support from the results of the current study and other studies done under the project.
However, in the case of the present study taken alone, we have mentioned in the methodological consideration that the transferability of the study results are limited to similar settings and that more studies are needed from the area involving multiple stakeholders. We have further added in response to whether the results are viable for planning interventions. The following lines are incorporated at the respective place:
“This is a preliminary study and our results’ transferability is limited to areas with similar settings. It's vital to note that our study does not include informal pharmacists, which is critical for future studies. Further research is needed involving multiple stakeholders, such as patients and informal pharmacists, to have a deeper knowledge of individual aspects at all levels of the model. This would aid in a more thorough understanding of and response to the factors that drive antibiotic prescriptions, guiding an effective antibiotic stewardship programme.”
The changes are incorporated in the section: Discussion; sub-section: Methodological considerations.
- Line 404: I recommend explaining the meaning of the acronym "FGD" at its first occurrence.
Response 8:
Thank you for noting it down, change has been added at the respective place.
- Line 408: Author contributions are already included in the paragraph "Author contribution", you could replace “So, the first author…”with "We contacted...". Do the same where you used the same methodology, for example line 422
Response 9:
Thank you for your suggestion. We would like to bring to your information that this paper is part of doctoral studies of the first author so to elaborate on her work contribution we choose to present contribution within the text in this manner which is commonly used in other papers as well.
- Line 450: Since the acronym "KAP" appears the first time at line 94, I recommend I recommend explaining its meaning at that point as well.
Response 10:
Thank you for noting it down, change has been added at the respective place.
- The article number 46 has already been published; the wording “in press” is incorrect.
Response 11:
Thank you for noting it down, change has been added at the respective place.
Yours Sincerely
Dr Shweta Khare
Department of Global Public Health
Karolinska Institutet
Email: shweta.khare@ki.se
Mar 16 2022
Round 2
Reviewer 2 Report
Thank you for having improved this noteworthy manuscript as suggested. There are no major flaws to point out.
Please, make sure to remove any layout problem from page 4.